# Comparison of Metabolome and Transcriptome of Flavonoid Biosynthesis Pathway in a Purple-Leaf Tea Germplasm *Jinmingzao* and a Green-Leaf Tea Germplasm *Huangdan* reveals Their Relationship with Genetic Mechanisms of Color Formation

**DOI:** 10.3390/ijms21114167

**Published:** 2020-06-11

**Authors:** Xuejin Chen, Pengjie Wang, Yucheng Zheng, Mengya Gu, Xinying Lin, Shuyan Wang, Shan Jin, Naixing Ye

**Affiliations:** College of Horticulture, Fujian Agriculture and Forestry University/Key Laboratory of Tea Science in University of Fujian Province, Fuzhou 350002, China; 1180311002@fafu.edu.cn (X.C.); 2180311002@fafu.edu.cn (P.W.); 1170311017@fafu.edu.cn (Y.Z.); 1190311005@fafu.edu.cn (M.G.); 1190311011@fafu.edu.cn (X.L.); 1100311012@fafu.edu.cn (S.W.)

**Keywords:** *Camellia sinensis*, leaf coloration, flavonoid biosynthesis, widely targeted metabolomic, transcriptomic, coexpression analysis

## Abstract

Purple-leaf tea is a phenotype with unique color because of its high anthocyanin content. The special flavor of purple-leaf tea is highly different from that of green-leaf tea, and its main ingredient is also of economic value. To probe the genetic mechanism of the phenotypic characteristics of tea leaf color, we conducted widely targeted metabolic and transcriptomic profiling. The metabolites in the flavonoid biosynthetic pathway of purple- and green-leaf tea were compared, and results showed that phenolic compounds, including phenolic acids, flavonoids, and tannins, accumulated in purple-leaf tea. The high expression of genes related to flavonoid biosynthesis (e.g., *PAL* and *LAR*) exhibits the specific expression of biosynthesis and the accumulation of these metabolites. Our result also shows that two Cs*UFGT*s were positively related to the accumulation of anthocyanin. Moreover, genes encoding transcription factors that regulate flavonoids were identified by coexpression analysis. These results may help to identify the metabolic factors that influence leaf color differentiation and provide reference for future research on leaf color biology and the genetic improvement of tea.

## 1. Introduction

Tea (*Camellia sinensis* (L.) O. Kuntze) is a perennial evergreen woody plant species that belongs to the Theaceae family of angiosperms [1]. Tea plants are grown widely in the tropical and subtropical zones around the world, including mainly China, Japan, India, and Kenya [2]. Purple-leaf tea, a novel germplasm of *C. sinensis*, has a unique color and a high content of bioactive flavonoids [3,4,5]. Flavonoids, as the most pronounced secondary metabolites in plants, play an important role in human health benefits and have potential physiological functions with antioxidant, antiaging, anticancer abilities and balance blood glucose. The intake of flavonoid composition in tea can reduce incidence of cardiovascular disease, cancer, diabetes, and hypertension [6,7].

Flavonoids, which belong to the phenolic class of compounds [8], include flavones, flavonoids, isoflavones, anthocyanins, flavanols, flavonols, and derivatives (e.g., catechins) [9,10]. Although flavonoids (including glycosyl derivatives) show a low threshold, they not only enhance the bitter effect of caffeine but also contribute to the astringency and bitterness of tea infusions [11]. Most genes have been actively studied for their contributions to flavonoid accumulation in many plants [12], including Arabidopsis (*Arabidopsis thaliana*) [13], maize (*Zea mays*) [14], figs (*Ficus carica* L.) [15], and tea plant (*Camellia sinensis*) [16]. The red color of cotton is mainly caused by flavonoid accumulation. Except for *4CL* and *F3′5′H* of flavonoid-related genes in red cotton, other genes that regulate flavonoid biosynthesis have higher transcription levels than those of white cotton [17]. The enzymatic properties related to flavonoid accumulation in vitro have been confirmed [18,19]. Under the regulation of the *CsFLS* gene, the content of flavonoids in the tea of the albino cultivar “*Rougui*” is higher than that of normal green-leaf cultivars [20]. However, the regulatory pattern of flavonoid biosynthesis in purple-leaf tea not only differs from that of other currently ornamental plants but also does not match those characterized for regulating flavonoid in *C. sinensis* [21].

In addition to the synthetic enzymes involved in the pathway, regulation by TFs has been elucidated. Several TFs that regulate the expression of the structural flavonoid genes have been identified in tea plants. For example, *MYB11*, *MYB12*, and *MYB111* play important roles in the biosynthesis of flavonol through the activation of upstream genes, such as *CHS*, *CHI*, and *F3H*, while the MYB-bHLH-WD40 (MBW) complex affects downstream genes that contribute to various branches, such as anthocyanins and procyanidin [22,23]. In tea plants, *CsMYB2* promotes the expression of *CsF3′H*, whereas *CsMYB26* is positively correlated with *CsLAR* expression [24]. *MYB* and *bHLH* exert regulatory effects on EGCG biosynthetic genes in plants [19]. However, the molecular mechanism of the transcription regulation of flavonoid biosynthesis in purple-leaf tea remains unknown, and the gene regulatory network of flavonoid biosynthesis has not been fully investigated.

Here, we utilized an integrated widely targeted metabolomic and transcriptomic approach to reveal flavonoid biosynthesis in purple- and green-leaf tea. Our results provide new insights into the 49 flavonoid compounds that were accumulated, including six anthocyanins from purple-leaf tea. Additionally, we characterized several differentially expressed genes (DEGs) that were significantly up- or downregulated in flavonoid biosynthetic structural genes and transcriptional regulatory genes of flavonoid biosynthesis. Some of these genes were further verified by qRT-PCR. These findings revealed that the analysis of metabolic expression profiles and molecular regulation levels enriched the understanding of purple-leaf tea and provided valuable theoretical support for further exploration of new health products of purple-leaf tea.

## 2. Results

### 2.1. Comparison of Morphological Phenotypes between JMZ and HD

Significant differences between JMZ and HD were observed in the morphological phenotypes, especially concerning leaf color (Figure 1). The leaf of the cultivar JMZ exhibits lustrous surfaces and sharply jagged edges compared with HD. Furthermore, compared with HD, the color of buds and leaves on the young shoots was different from that of the old leaves in JMZ plant. The buds and young leaves were purple, but the old leaves on the lower part of the tea plant were green.

### 2.2. Metabolite Difference between JMZ and HD

To compare the metabolite difference of purple-leaf tea JMZ and green-leaf tea HD, dataset obtained from LC-ESI-MS/MS were subjected to PCA and PLS-DA analyses. The results revealed that purple-leaf tea JMZ and green-leaf tea HD clearly separated in score plots (Figure 2a,b). Furthermore, volcano plots were generated to display the significant differences between JMZ and HD (Figure 2c). In the present work, 435 metabolites were identified in purple-leaf tea JMZ and green-leaf tea HD. Among the metabolites, 127 differentially expressed metabolites (SCMs) were detected between purple-leaf tea JMZ and green-leaf tea HD, including 57 downregulated and 70 upregulated metabolites, respectively (Figure 2d). Many flavonoids and anthocyanins were upregulated in the purple-leaf tea JMZ. Moreover, some metabolites, including phenolic acids, amino acids, and derivatives, were downregulated.

To further compare the fold changes in SCM levels of purple-leaf tea JMZ and green-leaf tea HD, we generated a heat map of the 127 SCMs by using TBtools (Figure 3 and Appendix A), marked with green to represent elevated levels of metabolites and red to represent decrease. Forty-nine flavonoids were found to be differentially accumulated in purple-leaf tea JMZ and green-leaf tea HD, of which in purple-leaf tea JMZ, 37 flavonoids, including cyanidin 3-*O*-glucoside, delphinidin 3-*O*-glucoside, peonidin 3-*O*-glucoside chloride, cyanidin 3-rutinoside, cyanidin 3-*O*-galactoside, sissotrin, and 2′-hydroxygenistein, were significantly increased, whereas 12 flavonoids, including syringaldehyde, chrysoberyl *O*-glucuronic acid, 6-hydroxykaempferol-7,6-*O*-diglucoside, and quercetin 3,7-bis-*O*-β-D-glucoside, were decreased. Compared with green-leaf tea HD, 22 of the 26 significantly changed phenolic acids, including isochlorogenic acid A, 3,4-dicaffeoylquinic acid, neochlorogenic acid, isochlorogenic acid C, cryptochlorogenic acid, and chlorogenic acid methyl ester, were decreased in JMZ, whereas cynarin, cinnamic acid, and 2-methoxybenzoic acid were increased. All significantly changed amino acids and derivatives showed markedly increased abundance in purple-leaf tea JMZ, especially with a 3.34-fold increase. Seven out of the 10 significantly changed nucleotides and derivatives, including allopurinol, xanthosine, xanthine, inosine 5’-monophosphate, and 3-methylxanthine, showed markedly higher abundances in purple-leaf tea JMZ than in green-leaf tea HD. Among the six significantly changed alkaloids, four compounds, namely, fer-agmatine, N-feruloyl agmatine, indole, and isoquinoline, showed markedly higher abundances in purple-leaf tea JMZ than in green-leaf tea HD. Interestingly, four significantly changed organic acids were markedly decreased in green-leaf tea HD.

### 2.3. Accumulation Patterns of Phenolic Acids and Tannins in JMZ and HD

Phenolic acids are aromatic secondary metabolites that are widely distributed in the plant kingdom [25]. Some phenolic acids such as gallic acid have attracted considerable attention because of their powerful antiradical, antioxidative, antitumor, and antimicrobial properties [26,27,28]. The heatmap identified 26 significantly different metabolites in leaves of the two different cultivars (Figure 3). The contents of most of the significant metabolites, except for chlorogenic acid methyl ester, cynarin, cinnamic acid, and 2-methoxybenzoic acid, were lower in purple-leaf tea JMZ than in green-leaf tea HD. Therefore, the phenolic acid contents may be negatively correlated with purple-leaf tea and positively correlated with green-leaf tea.

Tannins are important compounds that enhance the color stability and the anthocyanin content [29]. Ten tannins, namely, theasinensin C (T1), 3,3′,4′,5,5′,7-hexahydroxyflavan-(4->8)-3,3′,4′,5,5′,7-hexahydroxyflavan (T2), 3,3′,4′,5,5′,7-hexahydroxyflavan-(4->8)-3,3′,4′,5,5′,7-hexahydroxyflavan (T3), 3,3′,4′,5,5′,7-hexahydroxyflavan-(4->8)-3,3′,4′,5,5′,7-hexahydroxyflavan (T4), 3,3′,4′,5,5′,7-hexahydroxyflavan-(4->8)-3,3′,4′,5,5′,7-hexahydroxyflavan (T5), 3,3′,4′,5,5′,7-hexahydroxyflavan-(4->8)-3,3′,4′,5,5′,7-hexahydroxyflavan (T6), theaflavin (T7), epicatechin-epiafzelechin (T8), tetragalloylglucose (T9), and procyanidin C2 (T10), showed significant discrepancies between purple-leaf tea JMZ and green-leaf tea (Figure 3). Nine of these tannins were significantly increased in purple-leaf tea. T3 and T5 showed 15.203- and 15.114-fold changes, respectively, in purple-leaf tea compared with green-leaf tea. However, only one metabolite in JMZ, namely, tetragalloylglucose (T9), was upaccumulated in tannin biosynthesis.

### 2.4. Differential Expression of Genes in JMZ and HD

To obtain a reference transcriptome for the purple-leaf and green-leaf tea, we constructed an RNA-seq library with the total RNA of tea leaf samples. A total of 37.98–49.55 million raw reads were obtained by using the Illumina Hi-Seq 4000 platform. After deleting reads containing adapters or poly-N and low-quality reads, high-quality data with a Q30 percentage of 90.03%–90.58% and GC percentage of 45.10%−45.58% were available for analysis (Table 1 and Appendix A). All the unigenes were successfully annotated through alignment to the reference database with 35,967,226 (93.18%), 34,512,682 (93.41%), 41,291,024 (93.06%), 44,416,160 (93.33%), 45,021,912 (93.64%), and 41,274,294 (93.99%). All transcriptome data sets were stored in the NCBI SRA database under the accession number PRJNA638184.

We annotated the GO and KEGG pathway functional enrichment analyses by using the Blast-GO and KEGG pathway databases, respectively. The DEGs were divided into three main categories, namely, molecular function, biological process, and cellular component, and 49 subcategories of the GO classification. In the group, 859, 648, and 213 unigenes were assigned to the biological process, molecular function, and cellular component terms. These genes were further classified into 48 functional subcategories based on mapped homology (Figure 4A). Genes in the biological process category were primarily matched and classified into metabolic processes, cellular process, and single-organism processes. In the molecular function term, most of the unigenes exhibited catalytic activity, binding, and transporter activity. The most abundant GO terms in the cellular component category included the membrane, cell, cell part, organelle, and organelle part. In the KEGG pathway-enrichment analysis, matches were found for 3916 unigenes, which were mapped to 116 KEGG pathways. According to the KEGG pathway database, the main enriched metabolic process was phenylpropanoid biosynthesis in JMZ and HD (Figure 4B).

### 2.5. Analysis of DEGs and SCMs Related to the Flavonoid Biosynthesis Pathway in JMZ and HD

Based on the flavonoid metabolite results, the pattern of differentially accumulated flavonoids was significant when the ratio >2 or ratio ≤1/2, q-value ≤ 0.05, and VIP ≥1 between purple- and green-leaf tea. Among the peaks that indicated differentially accumulated metabolites, 49 were attributed to flavonoids.

Flavonoids are synthesized via the phenylpropanoid pathway (Figure 5) through the transformation of phenylalanine into 4-coumarol-CoA, which ultimately enters the flavonoid biosynthesis pathway. Enrichment analysis revealed that 17 genes were involved in the flavonoid biosynthesis pathway in purple-leaf tea JMZ [30]. However, we only detected *PAL*, *C4H*, *4CL*, *DFR*, *LAR*, *ANS*, and *UFGT*, whereas *CHS*, *CHI*, *F3H*, *F3′5′H*, and *F3′H* were not detected. We observed phenylalanine, cinnamic acid, p-coumaric acid, delphinidin-3-glucoside, cyanidin 3-rutinoside, cyanidin 3-*O*-galactoside, and cyanidin 3-*O*-glucoside by widely targeted metabolomics. Significant differences were observed in the expression of enzymes detected between the two different colored tea cultivars. Interestingly, the gene downstream in the flavonoid biosynthesis pathway differed most significantly among the different tea cultivars. For example, in purple-leaf tea JMZ, the principal DEGs were *TEA024758*, *TEA023829*, and *TEA015762*, which encode *DFR* and *ANS*. Genes involved in *LAR* and *UFGT* presented high expression levels in purple-leaf tea JMZ. The final step in the anthocyanin biosynthesis pathway is 3-glucoside formation by uridine 5′-diphosphate (UDP)-glucose: *UFGT*. The expression of the *UFGT* gene was active in the last step of anthocyanin modifications. The existence of anthocyanins and the generation of purple pigmentation in cells were stable because of the role of the abovementioned genes [31].

### 2.6. Confirmation of Flavonoid Regulatory Genes by Using qRT-PCR

To further validate our RNA-seq expression profile data, we performed qRT-PCR assays on 17 selected structural genes, including one *PAL*, two *C4H*, three *4CL*, three *CHI*, one *F3′5′H*, one F3′H, one *F3H*, two *DRF*, one *LODX/ANS*, and two *UFGT.* With respect to the purple-leaf tea, 6/17 (35%) were found to be inconsistent with respect to expression between the qRT-PCR and RNA-Seq data. Overall, 11/17 (65%) were consistent with respect to expression between the qRT-PCR and RNA-Seq data. Those included *CsPAL*, *CsC4H-1*, *Cs4CL*, *Cs4CL-2*, *CsCHI-1*, *CsCHI-2*, *CsF3′5′H*, *CsF3′H*, *CsDFR-1*, *CsUFGT-1*, and *CsUFGT-2.* Detailed information of the main flavonoids-related up- and downregulated genes is presented in Figure 6.

### 2.7. Conjoint Analysis between Transcripts and Flavonoid Derivative

The inferred networks were confirmed by Pearson correlation analysis based on the data from metabolism, transcriptome, and transcription factors in the flavonoid biosynthesis of purple-leaf tea JMZ. Eleven structural genes in the phenylpropanoid-flavonoid pathways, including *PAL*, *C4H*, *4CL*, *DFR*, *LAR*, *ANS,* and *UFGT,* were the main regulators. We further analyzed the transcription factor families involving MYB, bHLH, and WD40, which perform crucial roles in flavonoid biosynthetic structural genes; among the detected transcription factors, 66 *MYBs* and 187 *bHLH* were identified. Structural genes in the phenylpropanoid–flavonoid pathways and two TFs (*MYB* and *bHLH*) were used to coexpress the network and identify the genes that may regulate flavonoid compounds (Figure 7). These results suggested that the flavonoid DEGs and TFs may contribute to flavonoid biosynthesis in DEMs.

To further understand the regulatory pattern of the structural genes involved in the flavonoid biosynthesis pathways by transcripts, we performed gene coexpression analysis. The correlations between the DEGs in the flavonoid pathway illustrated the metabolites in the pathway (Figure 7A). According to our dataset, the networks revealed the presence of a hub gene, namely, “shikimate *O*-hydroxycinnamoyltrasferase” (*CsHCT*, *TEA029054*), in purple-leaf tea JMZ. The expression level of *CsHCT* was identified that was positively related to the contents of flavonoids, dihydroflavone, isoflavones, chalcones, anthocyanins, flavanols, dihydroflavonol, and flavonoid carbonoside in purple-leaf tea JMZ. Therefore, the results indicated that *CsHCT* is a key regulator of the flavonoids biosynthesis in purple-leaf tea JMZ. Furthermore, we also obtained the transcription factor that may be involved in the regulation of flavonoid biosynthesis. Correlation calculations between these identified differentially expressed *MYB* and *bHLH* TFs and DEMs were conducted via coexpression analysis. The results showed that 45 MYB TF expressed a high correlation with four flavonoids (flavonoid carbonoside, flavanols, dihydroflavonol, and anthocyanins), whereas 51 *bHLH* TF expressed a high correlation with nine flavonoids (flavonoids, dihydroflavone, isoflavones, chalcones, anthocyanins, flavanols, flavonols, dihydroflavonol, and flavonoid carbonoside) (*r* > 0.9, Figure 7B,C). Notably, zinc finger A20 (*TEA016255*), GPI-anchored protein (*TEA032923*), and F-box/LRR-repeat protein (*TEA000314*), which are group members of the R2R3-MYB TFs, were the key dominant factors of flavonoid biosynthesis and closely positively related to the four flavonoids (anthocyanins, flavanols, dihydroflavonol, and flavonoid carbonoside). The genes, namely, *bHLH121* (*TEA019255*), *MRS2-F* (*TEA008908*), *actin-depolymerizing factor* (*TEA020752*), and *MOS14* (*TEA025704*) had the highest degree of connectivity with flavonoid synthesis metabolites. Thus, *CsHTS*, *MYB*, and *bHLH* were identified as key genes for flavonoid synthesis.

### 2.8. Catechin Content in JMZ and HD

Catechins are responsible for astringency and bitterness [8]. A schematic diagram of catechin biosynthesis is shown in Figure 8. The content of C, EC, EGCG, EGC, GC, and ECG in JIM and HD were compared (Table 2). The EC, EGCG, EGC, GC, and ECG content were significantly higher in purple-leaf tea (JMZ) than those in the control leaves (HD). The results showed that the EC, EGCG, and ECG were predominant catechin compounds in JMZ, and their contents reached to 18.58, 55.17, and 22.88 mg/g, respectively. The total catechin amounts in JMZ were significantly (1.4 folds) higher than those in HD.

## 3. Discussion

The unique and delightful leaf color of tea is mediated by specific metabolic compounds and gene expression. The association of metabolite abundances and gene expression has been attracted increasing interest in recent years. In the present study, integrative analysis of metabolome and transcriptome profiles was performed to elucidate the mechanism of color formation in JMZ, a novel germplasm in China. The extensive metabolome datasets of JMZ were obtained for the first time in this study, which could provide a theoretical base for further study on mechanism of formation of pigments and genetic improvement of purple-leaf tea germplasm.

The complex network of metabolite profiling is remarkably changed in the diverse color of tea leaves [33]. The transcriptome and metabolome results of purple-leaf tea (JMZ) and green-leaf tea (HD) suggested that different molecular programs are involved in metabolites biosynthesis. In *C. sinensis* cv. *Zixin,* high levels of bioactive compounds, e.g., flavonoid, chlorophyll, and carotenoid, are important for the formation of purple-leaf coloration [4]. In the current research, six anthocyanin components were detected in young shoots, including delphinidin 3-*O*-glucoside, cyanidin 3-*O*-glucoside, peonidin 3-*O*-glucoside chloride, cyanidin 3-*O*-galactoside, cyanidin 3-rutinoside, and cyanidin chloride. Among them, the contents of peonidin 3-*O*-glucoside chloride, cyanidin 3-*O*-glucoside, and delphinidin 3-*O*-glucoside in JMZ were significantly higher than those in HD, which were 13.39, 6.78, and 6.73 times higher, respectively. The main concern is composition of anthocyanin 3-*O*-glucoside in the cultivars of purple-leaf tea, that agreed with the previous reports of Zhou et al. [34] who observed that the anthocyanin 3-*O*-glucoside composition was mainly differentially accumulated at developmental stages in various colored tea flowers. Moreover, “*Sunrouge*,” as one of the anthocyanin-rich teas in Japan, has identified four anthocyanins, including delphinidin-3-*O*-β-D-(6-(E)-p-coumaroyl)glucopyranoside, cyanidin-3-*O*-β-D-(6-(E)-p-coumaroyl)glucopyranoside, delphinidin-3-*O*-β-D-(6-(E)-pcoumaroyl) galactopyranoside, and cyanidin-3-*O*-β-D-(6-(E)-pcoumaroyl) galactopyranoside [35]. We also found other differentiation of phenolic compounds. In particular, many catechins, including EC, EGCG, EGC, GC, and ECG, were significantly increased in JMZ. The diversity and abundance of catechins are involved in the characteristic taste of some tea cultivars [36]. For instance, 70%−75% of the bitterness and astringency of green tea are related to these catechins [8]. Briefly, presence of most of the significantly changed flavonoids, anthocyanins, and catechins contents in the purple-leaf tea (JMZ) suggests that it may have high application value.

Previous studies have identified some key structural genes from the flavonoid pathways regulating coloration [37]. Our data showed six differentially expressed genes in JMZ, including *PAL*, *C4H*, *4CL*, *DFR*, *ANS*, and *UFGT* genes. *PAL* is as an important modulate pinpoint that regulates the flavonoid accumulation in the *C. sinensis* cv. *Zijuan* [38]. In agreement with other results, *PAL* was proposed as a pivotal gene responsible for regulating the purple pigmentation in JMZ. Furthermore, *C4H*, *4CL*, *DFR*, and *ANS* are the key branch-point genes that regulate the accumulation of flavonoid [39]. Our studies showed that low expression of *C4H*, *4CL*, *DFR,* and *ANS* may have a negative correlation to total anthocyanins in purple-leaf tea JMZ. Previous studies showed that *PAL, C4H, and ANS* genes were upregulated in the *C. sinensis* cv. *Foshou*, which may contribute to the formation of anthocyanin compounds [40]. Besides, *UFGT* is the last enzyme in anthocyanin biosynthetic pathway, and it is the key to anthocyanin stability and water solubility [41]. The expression of *UFGT* genes were detected in red grape cultivars, but not in white cultivars [42]. A similar result was reported by Yonekura-Sakakibara et al. (2008) for *Arabidopsis* [43]. They found that flavonoid 3-O-glucosyltransferase (*AtUDT78D2*) catalyzes the glycosylation of both flavonol and anthocyanin, and the Arabidopsis *UGT78D2* mutant exhibits a high anthocyanin level. Interestingly, we investigated the UFGT-like gene expressed in JMZ and isolated two genes, namely, *CsUFGT1* (*TEA004632*) and *CsUFGT2* (*TEA004632*), which are associated with anthocyanin accumulation in purple-leaf tea.

Our results confirmed that a novel gene, *HCT,* was characterized by interaction analysis of metabolism and different genes of flavonoid biosynthesis pathway. Although some *HCT* genes were studied, their function is still unclear. The expression of *HCT* displayed that the expression of *HCT* gene was repressed on the lignin biosynthesis of *Arabidopsis* [44], and *HCT* positively regulates flavone and flavonol biosynthesis in *Euscaphis konishii* Hayata [45]. In the present study, the expression of *CsHCT* positively related to the contents of flavonoids, dihydroflavone, isoflavones, chalcones, anthocyanins, flavanols, dihydroflavonol, and flavonoid carbonoside in purple-leaf tea JMZ. Therefore, these results demonstrated that *HCT* may be a key modulator to promote the flavonoid biosynthesis in plants. However, the relationship between *HCT and* the catalysis formation of flavonoids needs for further verification.

Furthermore, the abundant TFs with regulatory function in flavonoid biosynthesis belong to R2R3-MYB and bHLH protein family [46]. A positive or negative correlation exists between MYB TFs, which are the key regulators encoding structural genes involved in flavonoid biosynthesis [47,48]. In Tartary buckwheat (*Fagopyrum tataricum*), the *AtMYB12* of MYB TFs members was the main regulator in the four flavanones [47]. The function and regulatory relationships between R2R3-MYB and the flavonoid biosynthesis have been discussed in crabapple [49]. In the present study, we identified three members of the subgroup R2R3-MYB TFs, namely, *Zinc finger A20* (*TEA016255*), *GPI-anchored protein* (*TEA032923*), and *F-box/LRR-repeat protein* (*TEA000314*), which are closely related to four flavonoids, namely, anthocyanins, flavanols, dihydroflavonol, and flavonoid carbonoside. The regulation of flavonoid biosynthesis involves different bHLH TFs targeting genes [50,51]. In *Arabidopsis*, *AtbHLH8* (PIF3) bound and activated the expression of anthocyanin synthesis-related genes, including *CHS*, *F3H*, *DFR*, and *ANS* [52]. In the present study, we elucidated that the expression levels of *bHLH121* (*TEA019255*), *MRS2-F* (*TEA008908*), *actin-depolymerizing factor* (*TEA020752*), and *MOS14* (*TEA025704*) were highly correlated with the contents of flavonoid synthesis metabolites.

## 4. Materials and Methods

### 4.1. Tea Plant Materials

Two four-year-old tea cultivars, a *C. sinensis* cv. *Jinmingzao* (JMZ) with purple-leaf and a *C. sinensis* cv. *Huangdan* (HD, a widely grown variety in China) with green-leaf tea, were cultivated on a tea plantation of Wuqu in Fuan city, Fujian Province, China (E119°66′ N27°10′; 150–200 m above sea level). For each cultivar, one bud and two young shoots were harvested from 24 individuals used for the metabolite analysis (3 biological replication × 4 individual for each replicate) and RNA-seq (3 biological replication × 4 individual for each replicate) on March 31, 2019. Samples were randomly collected from different branches of tea plants of each cultivar. All materials were frozen immediately in liquid nitrogen and then stored at −80 °C until further analysis. The DEGs and metabolites of purple-leaf and green-leaf tea were identified by transcriptome and widely targeted metabolomics, and functional genes with significant differences were verified by qRT-PCR. Six libraries, namely, JMZ_1, JMZ_2, JMZ_3, HD_1, HD_2, and HD_3 was constructed during the experiment.

### 4.2. LC-ESI-MS/MS Analysis and Differential Metabolite Identification

After freezing the tea samples with liquid nitrogen, a mixer-mill with zirconia (Zr; Z = 40) beads (MM 400, Retach) was used to grind the samples at 30 Hz for 90 s until they were powdered. Precisely, 100 mg of powdered sample was weighed and dissolved in 1.0 mL of 70% methanol extraction solution. The dissolved sample was stored overnight in a refrigerator at 4 °C. Three vortices were used to improve the extraction rate. The mixture (5 µL) was injected into an LC-ESI-MS/MS (LC, Shim-pack UFLC Shimadzu CBM30A system; ESI, MS, Applied Biosystems 6500 *Q*TRAP^®^) system. The detection was carried out as previously described [53]. Chromatographic separation was executed by an ACQUITY UPLC HSS T3 C18 (1.8 µm, 2.1 mm × 100 mm; Waters). The mobile phase consisted of A containing 0.04% acetic acid in deionized water and B containing acetonitrile with 0.04% acetic acid in acetonitrile. The A: B (*v/v*) elution profile was as follows: 0–11 min, 95A:5B; 11–12 min, 5A:95B; 12.1–15 min, and 95A:5B. The flow rate was maintained at 0.4 mL/min, and the column temperature was 40 °C.

The concentrations of metabolites were identified based on a widely targeted metabolomic method of Wuhan Maiteville Biotechnology Co., Ltd. (Wuhan, China). This method has been described in a study on tomatoes [54]. Metabolite identification was based on the parametric values (*m*/*z* data, retention time, and fragmentation partners) and compared with the self-built database (MetaWare) for annotation results. Principal component analysis (PCA) and partial least-squares discriminant analysis (PLA-DA) were conducted to identify the significantly changed metabolites (SCMs), where fold change ≥ 2 or fold change ≤ 0.5 and the metabolites with VIP ≥ 1. PCA and PLS-DA were implemented in R software (www.rproject.org/) to investigate tea metabolite variety-specific accumulation [55].

### 4.3. RNA-Seq and Data Processing

RNA isolation and purification from JMZ and HD, as well as cDNA library construction and sequencing, were performed as previously described [56]. Total RNA was extracted from the young shoots of JMZ and HD by using the pBIzol kit (BIOFLUX, Hangzhou Bori Technology, Hangzhou, China). RNA integrity of samples was measured using a Nano-Drop ultraviolet spectrophotometer (Thermo, Waltham, MA, USA) and a Bioanalyzer 2100 System (Agilent, Santa Clara, CA, USA). About 3 μg of RNA per sample was prepared, and sequencing was carried out on an Illumina HiSeq 4000 platform to generate 150 bp paired end reads for each sample. Two test samples were used to construct the transcriptome library and Illumina sequencing in Allwegene Biotechnology Co., Ltd. (Beijing, China).

### 4.4. Analysis of DEGs

Raw reads in FASTQ format were first processed by filtering out adapters and low-quality sequences. Clean reads were obtained by removing reads containing adapter and poly-N and low-quality reads from raw data. All clean reads were aligned to the reference transcripts of *C. sinensis* (http://tpia.teaplant.org/index.html) by applying TopHat2. Gene expression levels were calculated based on the number of reads mapped to the reference sequence to receive the unigenes by using HISAT2 and StringTie software. To assess gene expression levels, we calculated the FPKM of each gene. DEGs that were related to the metabolism of flavonoids (e.g., flavones, anthocyanidins, and chalcones) were functionally annotated by using GO and KEGG [57]. DEGs between JMZ and HD were enrolled based on threshold values of|log_2_(fold change)|≥ 1 and adjusted *p* ≤ 0.005.

### 4.5. qRT-PCR Validation of RNA-Seq Results

In total, 17 genes were selected for qRT-PCR analysis to examine the different expression profiles. qRT-PCR detection system of the samples was performed as previously described [58]. The relative expression of each gene was calculated after normalization to that of the *CsGAPDH* (GE651107) gene. The samples of the three independent biological replicates were used for the analyses. The expression levels of candidate genes were determined by applying the 2^−△△*C*T^ method [59]. The specific primers used for RT-qPCR are listed in Appendix A.

### 4.6. UPLC-TOF-MS for Determination of Catechins Content

To further supplement the conclusion of metabolomics analysis, the content of catechins was analyzed by UPLC-TOF-MS. The analysis method adopts the method of Yue et al. [60], and we have made some modifications. Briefly, 0.05 g tea powder was weighed, placed into a 5 mL glass tube, and added to 5 mL of extraction solution (methanol: water = 7:3). The solution was subjected to ultrasonic extraction for 30 min (ds-8510 DTH, Shanghai, China), centrifuged for 3 min (Avanti-e, Beckman Coulter, USA), and tested on the machine after passing through a 0.22 µm PTFE filter membrane (f2513-4, Thermo Scientific, USA). The UPLC TSS T3 ultrahigh performance liquid chromatography column (150 mm × 2.1 mm, 18 µm, Waters, the UK) was utilized with column temperature of 40 °C, injection amount of 5 μL, flow rate of 0.3 mL/min, mobile phase A comprising 0.1% formic acid water, and mobile phase B comprising acetonitrile, and catechin was identified by the following standards: ESI ionization mode, capillary voltage of 2.5 KV, sampling cone voltage of 50 V, ion source temperature of 120 °C, dissolvent gas temperature of 40 °C, cone gas of 50 L/h, dissolvent gas flow rate of 800 L/h, MS adopted the MS^e^ scanning mode, mass spectrum, scanning range of 100–1000 *m*/*z*, scanning rate of 0.1 s, centroid data format, low collision energy of 5 V, and high collision energy of 20–40 V.

## 5. Conclusions

The formation mechanisms of leaf coloration in purple-leaf tea were investigated using widely targeted metabolomic and transcriptomic approaches. The accumulation of flavonoid metabolites, specifically, the high accumulation of anthocyanin and catechin compounds in purple-leaf tea, was observed. In addition, the related forms of the regulation of the structural genes involved in the flavonoid biosynthesis pathway were investigated. The involvement of these structural genes or their regulators (i.e., TFs) may provide a mechanism to improve the accumulation of red anthocyanins in purple-leaf tea, thereby maintaining beneficial metabolites during the purple-leaf period.

## Figures and Tables

**Figure 1 ijms-21-04167-f001:**
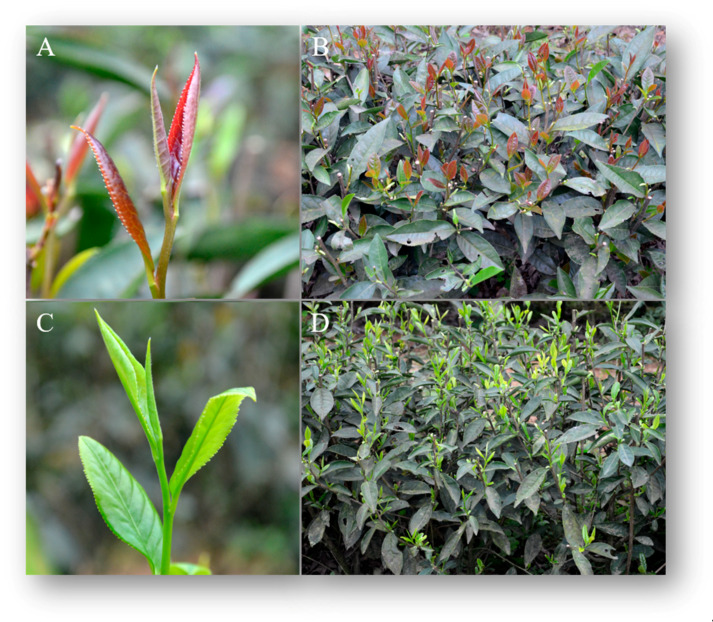
The morphological phenotypes of the two tea cultivars. (**A**) Young shoot of *C. sinensis* cv. *Jinmingzao* (JMZ). (**B**) Purple leaves of JMZ plant. (**C**) Young shoot of *C. sinensis* cv. *Huangdan* (HD). (**D**) Green leaves of HD plant.

**Figure 2 ijms-21-04167-f002:**
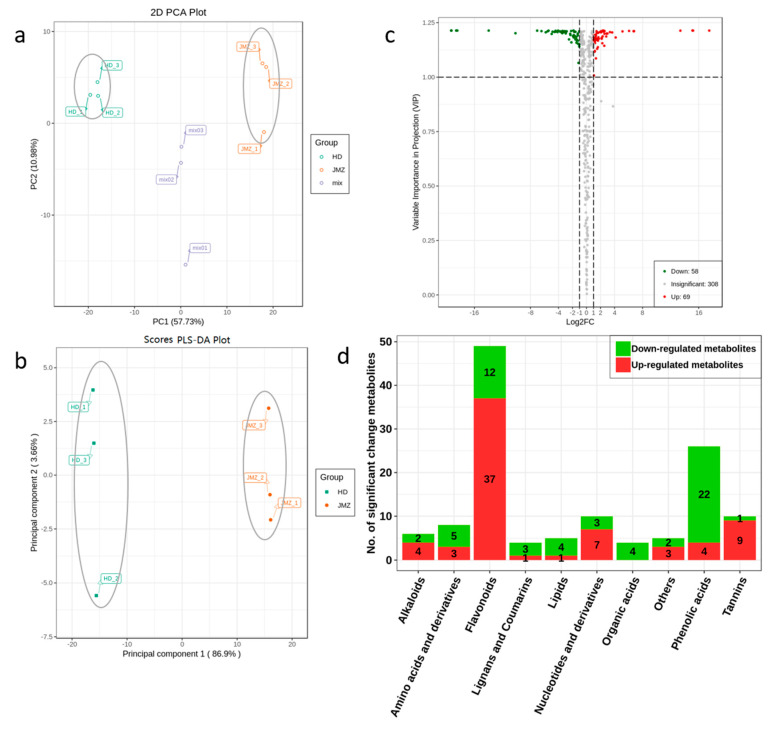
Multivariate statistical analysis of metabolites from *C. sinensis* cv. *Jinmingzao* (JMZ) and *C. sinensis* cv. *Huangdan* (HD). (**a**) PCA score plot of metabolites of the young shoots of JMZ (red) and HD (green); the *x*-axis represents the first principal component and the *y*-axis represents the second principal component. (**b**) PLS-DA score plot of metabolites between the young shoots of JMZ (red) and HD (green). (**c**) Volcano plots of metabolites between the young shoots of JMZ and HD. Metabolites with *q*-value > 0.05 are highlighted in red for upregulation and in green for downregulation; and (**d**) The 127 SCMs were divided into 10 categories.

**Figure 3 ijms-21-04167-f003:**
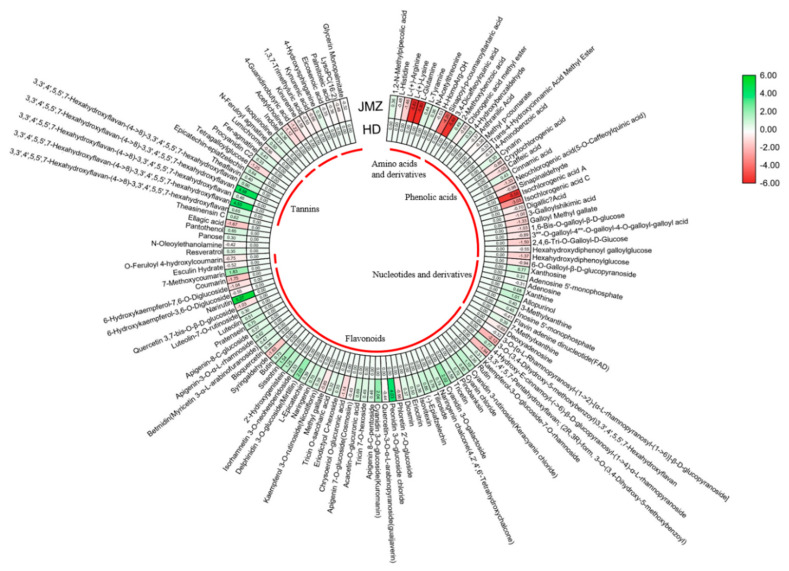
Heatmap of the fold change of significantly changed metabolites (SCMs) between the young leaves of *C. sinensis* cv. *Jinmingzao* (JMZ) and *C. sinensis* cv. *Huangdan* (HD). The color bar represents the normalized fold change values. Five categories with elevated SCM levels, including amino acids and derivatives, phenolic acids, nucleotides and derivatives, flavonoids, and tannins, are marked.

**Figure 4 ijms-21-04167-f004:**
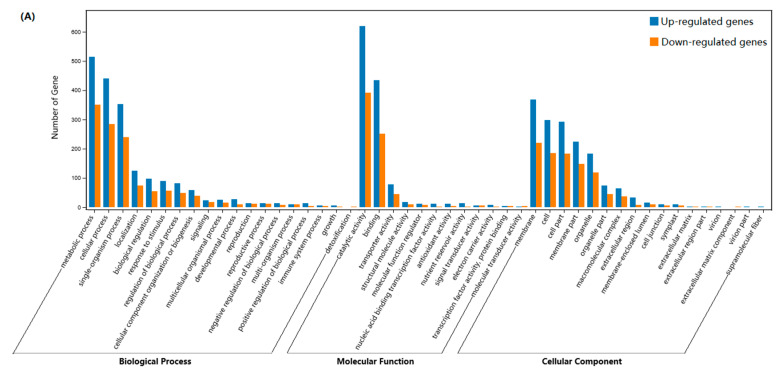
GO classification and KEGG pathway enrichment of differentially expressed genes in JMZ and HD. (**A**) The GO terms were classified into three categories: molecular function, cellular component, and biological process categories. The top 48 enriched GO terms were exhibited in each cluster. (**B**) Top 20 enriched KEGG pathway enrichment of differentially expressed genes (DEGs) of tea cultivar between JMZ and HD. The significantly changed pathways (Q-value < 0.05) are related to phenylpropanoid biosynthesis.

**Figure 5 ijms-21-04167-f005:**
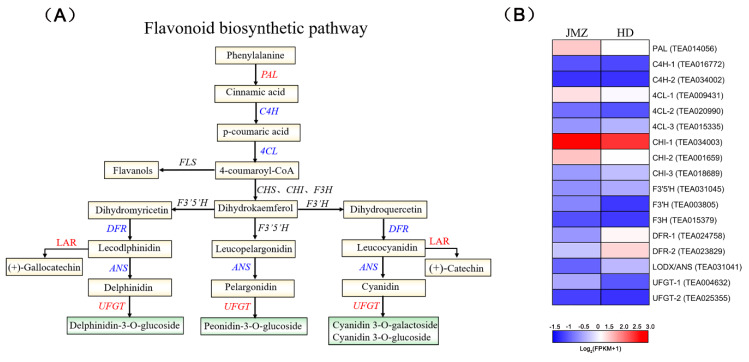
Transcript profiling of genes in the phenylpropanoid and flavonoid biosynthetic pathways in *C. sinensis* cv. *Jinmingzao* (JMZ) and *C. sinensis* cv. *Huangdan* (HD). (**A**) Flavonoid biosynthetic pathway. Proposed pathway of biosynthesis derived from the literature [30]. *PAL*, phenylalanine ammonia-lyase; *C4H*, cinnamic acid 4-hydroxylase; *4CL*, 4-coumarate CoA ligase; *CHS*, chalcone synthase; *CHI*, chalcone isomerase; *F3H*, flavanone 3-hydroxylase; *F3′H*, flavonoid 3′-hydroxylase; *DFR*, dihydroflavonol flavonol synthesis; *ANS/LDOX*, anthocyanidin synthase/leucocyanidin oxygenase; *LAR*, leucocyanidin reductase; *UFGT*, UDP glucose-flavonoid 3-o-glcosyl-transferase. (**B**) Heat map representation of the expression patterns of flavonoid-related genes. Gene expression is displayed as heat map depicting the log2 (FPKM) values. Red and blue font indicate up- and downregulated genes, respectively

**Figure 6 ijms-21-04167-f006:**
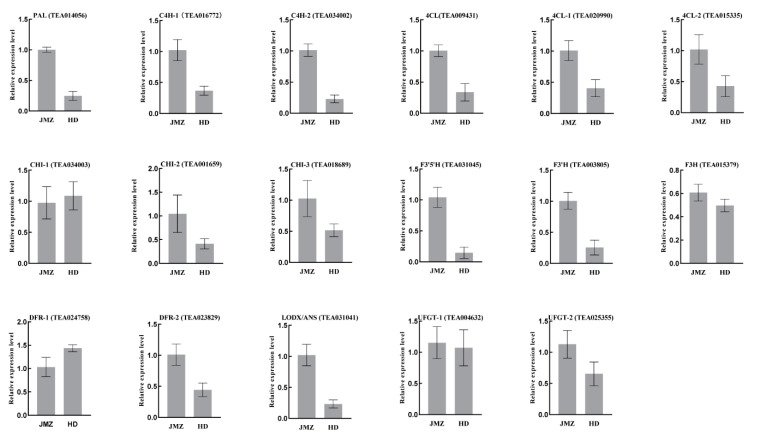
Expression of 17 selected candidate genes in purple-leaf and green-leaf tea by qRT-PCR validation. *PAL*, phenylalanine ammonia-lyase; *C4H*, cinnamic acid hydroxylase; *4CL*, coumadin CoA ligase; *CHS*, chalcone synthase; *CHI*, chalcone isomerase; *F3H*, flavonoid 3-hydroxylase; *F3′H*, flavonoid 3′-hydroxylase; *F3′5′H*, flavonoid 3′ 5′-hydroxylase; *DRF*, dihydroflavonol 4-reductase; *LAR*, leucoanthocyanidin reductase; *UFGT*, UDP glucose-flavonoid 3-o-glcosyl-transferase, and *SCPL1A*, type 1A serine carboxypeptidase-like acyltransferases. The *x*-axis represents two different tea cultivars of “JMZ” and “HD,” and the *y*-axis represents relative expression. The data represent the mean from three replicates with three biological repeats. Error bars indicate SE.

**Figure 7 ijms-21-04167-f007:**
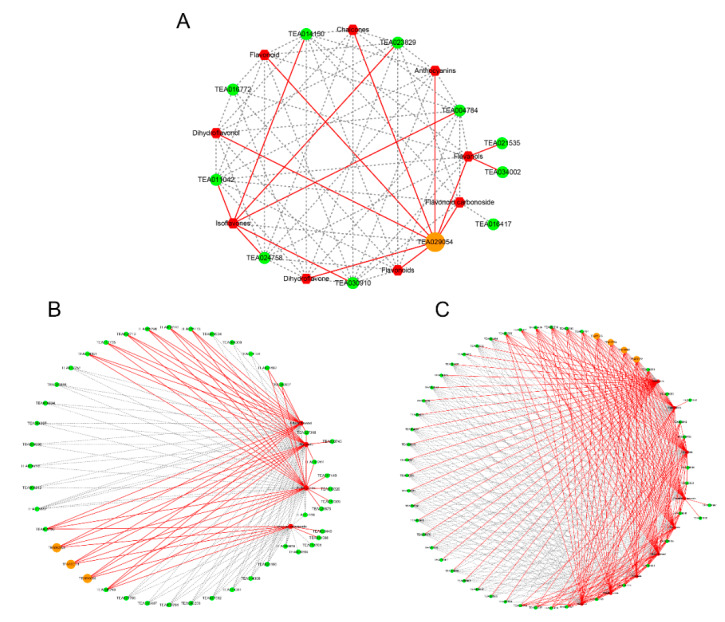
Connection network between flavonoid biosynthesis-related DEGs and differential accumulated flavonoid metabolites in purple-leaf tea JMZ. (**A**) Connection network between flavonoid biosynthesis structural genes and flavonoid metabolites, (**B**) connection network between *MYB* transcription factor genes and flavonoid metabolites, and (**C**) connection network between *bHLH* transcription factor genes and flavonoid metabolites. The green circle represents the key regulator of flavone and flavonol biosynthesis.

**Figure 8 ijms-21-04167-f008:**
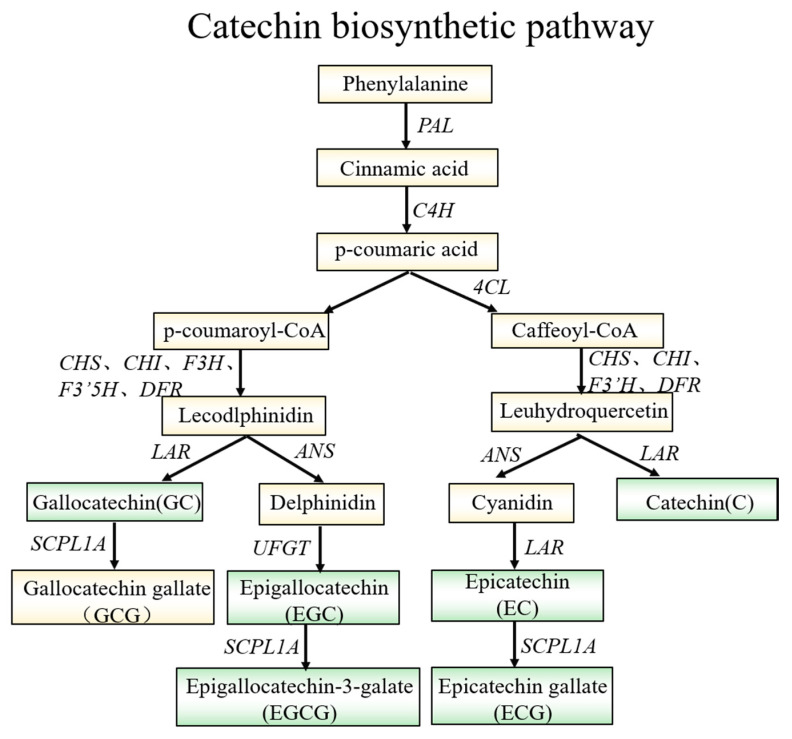
Catechin biosynthesis pathway. Proposed pathway of biosynthesis derived from the literature [32]. *PAL*, phenylalanine ammonia-lyase; *C4H*, cinnamic acid hydroxylase; *4CL*, coumadin CoA ligase; *CHS*, chalcone synthase; *CHI*, chalcone isomerase; *F3H*, flavonoid 3-hydroxylase; *F3′H*, flavonoid 3′-hydroxylase; *F3′5′H*, flavonoid 3′ 5′-hydroxylase; *DRF*, dihydroflavonol 4-reductase; *LAR*, leucoanthocyanidin reductase; *UFGT*, UDP glucose-flavonoid 3-o-glcosyl-transferase, and *SCPL1A*, type 1A serine carboxypeptidase-like acyltransferases.

**Table 1 ijms-21-04167-t001:** Quality of the transcriptomes of young shoots between *C. sinensis* cv. *Jinmingzao* (JMZ) and *C. sinensis* cv. *Huangdan* (HD).

Sample	Raw Reads	Clean Reads	Q30 (%)	GC (%)	Mapped Reads
JMZ_1	39848584	38598850	90.48	45.51	35967226 (93.18%)
JMZ_2	37983438	36946120	90.23	45.58	34512682 (93.41%)
JMZ_3	45639484	44371332	90.38	45.50	41291024 (93.06%)
HD_1	49081330	47588286	90.05	45.24	44416160 (93.33%)
HD_2	49552984	48077760	90.58	45.10	45021912 (93.64%)
HD_3	45187036	43911796	90.03	45.31	41274294 (93.99%)

**Table 2 ijms-21-04167-t002:** Content of catechins in JMZ and HD. All data are shown as the mean ± SE. (* means significant difference, *p* < 0.05).

Tea Cultivar	Catechins Content (mg/g)						Total Content (mg/g)
	C	EC	EGCG	EGC	GC	ECG	
JMZ	2.79 ± 0.02	18.58 ± 5.66 *	55.17 ± 0.41 *	38.76 ± 15.81 *	2.59 ± 0.64 *	22.88 ± 1.34 *	140.76 ± 23.88
HD	2.79 ± 0.025	6.77 ± 0.08	56.30 ± 0.51	5.82 ± 0.04	1.47 ± 0.05	26.24 ± 0.42	99.39 ± 1.12

(+)-catechin (C), (−)-epicatechin (EC), (−)-epigallocatechin gallate (EGCG), (−)-epigallocatechin (EGC), (−)-gallocatechin (GC), (−)-epicatechin gallate (ECG).

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
