# Peer review of "Comparison of Metabolome and Transcriptome of Flavonoid Biosynthesis Pathway in a Purple-Leaf Tea Germplasm Jinmingzao and a Green-Leaf Tea Germplasm Huangdan reveals Their Relationship with Genetic Mechanisms of Color Formation"

_ijms, 2020, doi:10.3390/ijms21114167_

Round 1

Reviewer 1 Report

The metabolome profile of a 70-% methanol extract of young leaves of a purple tea is compare to leaves from a green tea. Fresh leaves are harvested only once in the field and the same leaves are also used for RNAseq analysis to identify differentially expressed genes between the two tea cultivars. The authors have access to a very large database / analytical platform of metabolites and hence identify and maybe quantify those which are known to the database. It does not allow identification of new metabolites. The RNAseq data are supported by RT-PCR of a number for well chosen genes in the flavonoid pathway. There are a number of recent papers cited on metabolites in tea, also by the authors and it is not clear what is presented here is new. The authors claims the general benefit of these metabolites to the human health but tea is prepared with hot water, and not 70-% methanol. So are the metabolome identified relevant? Also will the profile be stable from harvest to harvest. 

There are some words missing in the manuscript which can be confusing or not understandable. Just some point to consider:

Title: it does not really reflect the content of the study, metabolite accumulation is not studied, but the metabolome is compared at a certain growth and developmental stage.

Abstract: what is new?

line 18 content.

line 19 "sensory quality", this paper does not present sensory data for the two tea cultivars!

line 19 "has major economic values" something is missing

line 20 "morphological diffentiation" this is not being studied

line 25 clarified the spatial patterns... how?

line 26 I am not sure you found a correlation, explain

line 27 what do you mean by "co-expression analysis"?

line 36 widely grown in China

line 39 largest secondayr metabolites; something is missing

line 47 figs

line 53-55  rephrase sentensse, what do you mean by "ornamental systems"?

line78 widely grown variety 

line 79 "tender shoots" other places you call it young leaves, please define the growth stage and be consistent

line 81 what is test samples? the "mix" in figure 2a?

line 88-89, be precise

line 131 technical replicates were for the analyses, please explain

line 156 here you call them cultivars other places varieties, be consistent

the legend should explain the figure wit sufficient detail, what is a HD plant etc.

line 160-161 rephrase sentense

legend to figure 2 could be more informative

legend to figure of JMZ and HD varieties

line 201 that are widely exist, rephrase

Figure 5, give reference to the pathway shown

Figure 6 expression of XX representative genes in the YY biosynthetic patheay of ....

line 303 This gene had corrections... I do not understand

line 307 with four and nine flavonoids, which ones?

line 322, the section on catechin content needs to be rewritten

Figure 8 reference is missing

Line 343-345 "fill the existing research gaps" how?

line 350 major flavonoid, rephrase

line 358 Six main type of plants occur, rephrase

line 361 Zijuan ???

line 367 absence of anthocyanin in HD, please elaborate and explain

line 369-370 rephrase

line 374. wrong the is not report on sonsory characteristics

line 433 "high accumulation of anthocyanins and catechins" there is not quantification data provided for the two tea varieties

Author Response

Dear Reviewer:

Thank you for your comments concerning our manuscript entitled “Comparision of metabolome and transcriptome of flavonoid biosynthesis pathway in a pueple-leaf tea germplasm Jinmingzao and a green-leaf tea germplasm Huangdan reveals the relationship with genetic mechanisms of color formation”. All of your comments are valuable and very helpful for revising and improving our manuscript, and we are also appreciated that your comments have given us a lot of inspiration and different ideas for our future research. We have studied comments carefully and have made corrections. Accordingly, we have uploaded the revised manuscript with all the changes highlighted by using the track changes mode in MS Word.

The corrections in the revised manuscript and the responds to the reviewer’s comments are as flowing:

Comments and Suggestions for Authors

The metabolome profile of a 70-% methanol extract of young leaves of a purple tea is compare to leaves from a green tea. Fresh leaves are harvested only once in the field and the same leaves are also used for RNA-seq analysis to identify differentially expressed genes between the two tea cultivars. The authors have access to a very large database / analytical platform of metabolites and hence identify and maybe quantify those which are known to the database. It does not allow identification of new metabolites. The RNA-seq data are supported by RT-PCR of a number for well chosen genes in the flavonoid pathway. There are a number of recent papers cited on metabolites in tea, also by the authors and it is not clear what is presented here is new. The authors claims the general benefit of these metabolites to the human health but tea is prepared with hot water, and not 70-% methanol. So are the metabolome identified relevant? Also will the profile be stable from harvest to harvest.

Response:The sampling was carried out on March 31, 2019. In China, the harvesting season of spring tea is from the end of March to the end of April. After the beginning of picking at the end of March, tea is picked about once a week until the end of April. After a winter, tea leaves are rich in biochemical components and nutrients, so spring tea is the best quality in a year. In most parts of China, only spring tea is harvested, and tea trees are pruned and maintained in other seasons. In addition, tender shoot with one bud and two leaves is the common picking requirement for most of Chinese tea. Therefore, the sampling time and tenderness of samples in this study are consistent with the practice of actual tea production in China.

Same to other articles (Wang et al., 2017; Wang et al., 2020), this study adopted the established and relatively mature method for the determination of tea metabonomics. Using 70% methanol to extract the biochemical substances of tea, the reason is that 96% of the chemical substances of tea are organic substances, using methanol to extract these organic substances, their extraction efficiency is higher, which can more truly reflect the differences of biochemical components between the two tea cultivars. If extracted with water, some important substances such as chlorophyll and carotenoids are insoluble in water, they will not be detected finally. The widely targeted metabonomics is performed in our study to comprehensively and preliminarily investigated the differences between Jinmingzao and green leaf tea cultivar. On this basis, we will screen and explore some interesting components and metabolic pathways, as the direction of further research in the later stage. Flavonoids and the metabolism are only one aspect of our concern. Therefore, we used the method of extracting tea with 70% methanol.

We really appreciate your very professional questions. Indeed, tea is brewed in boiling water, the ingredients in the tea soup that people drink are good for human health. This is also why our laboratory is presently carrying out a study on the metabonomics determination method of tea. This method is to extract the secondary metabolites of tea with boiling water for the metabolomics study. Although the categories and the response intensity of substances are far lower than those in this study, we still thought this is a very valuable study. The research of the new extraction method is in progress.

In this study, Jinmingzao is a new purple leaf tea germplasm. At present, there is few research report about Jinmingzao. We got a relatively comprehensive, preliminary and basic data by using metabonomics and transcriptome. At present, we want to explore Jinmingzao’s cultivar characteristics, physiological and biochemical characteristics, and then, we will pay attention to its health characteristics. In addition, drinking tea is not the only way to obtain tea beneficial ingredients. Eating tea is also a way, such as adding dried tea powder directly to food, such as Matcha cake, tea lozenges, etc.

Wang, Z.R.; Cui, Y.Y.; Vainstein, A, Chen S.W.; Ma H.Q.; Regulation of Fig (Ficus carica L.) Fruit Color: Metabolomic and Transcriptomic Analyses of the Flavonoid Biosynthetic Pathway. Front Plant Sci. 2017, 8:1990.

Wang, P.J.; Zheng, Y.C.; Guo, Y.C.; Liu, B.S.; Jin, S.; Liu, S.Z.; Zhao, F.; Chen, X.J.; Sun, Y.; Yang, J.F.; Ye, N.X. Widely targeted metabolomic and transcriptomic analyses of novel albino tea mutant of “Rougui”. Forests. 2020, 11, 229.

Point 1: Title: it does not really reflect the content of the study, metabolite accumulation is not studied, but the metabolome is compared at a certain growth and developmental stage

Response: Thank you for the title suggested. After thought about your suggestion, we changed the title to “Comparision of metabolome and transcriptome of flavonoid biosynthesis pathway in a pueple-leaf tea germplasm Jinmingzao and a green-leaf tea germplasm Huangdan reveals the relationship with genetic mechanisms of color formation”.

Point 2: Abstract: what is new?

Response: In our study, new tea cultivar with high anthocyanin content in leaves were found. We utilized an integrated widely targeted metabolomic and transcriptomic approach to reveal flavonoid biosynthesis in purple-leaf tea. The findings revealed that the analysis of metabolic expression profiles and molecular regulation levels enriched the understanding of purple-leaf tea, and provided valuable theoretical support for further exploration of new health products of purple-leaf tea.

Point 3: line 18 content.

Response: We are grateful to the reviewer for your meticulous inspections and we have modified to “content”" in line 18.

Point 4: line 19 "sensory quality", this paper does not present sensory data for the two tea cultivar

Response: Thanks for the Reviewer’s kind advice. It is generally believed that the quality and flavor of purple-leaf tea are remarkably different from that of green leaf tea (Shen et al., 2018; Zhao et al., 2009; Wang et al., 2003). At present, there are also many studies on improving the palatability of purple-leaf tea products. For example, according to the fact that anthocyanin is an important factor affecting the bitter taste of purple-leaf tea higher than green-leaf tea, a small amount of casein can be added in the processing of purple leaf tea to improve the taste of tea soup; In the process of brewing purple-leaf tea, licorice, stevia, honey and other foods can also be used as auxiliary materials to make healthy tea beverage with fresh export feeling.

Pan, Y.Y.; Wu, H.L.; Li, J.X.; Yang, C.W.; Liu, J. Advances in research and utilization of purple tea. Guangdong Agricultural Sciences. 2015, 42:8-12+17.

Shen, J.Z.; Zou, Z.W.; Zhang, X.Z.; Zhou, L.; Wang, Y.H.; Fang, W.P.; Zhu, X.J. Metabolic analyses reveal different mechanisms of leaf color change in two purple-leaf tea plant (Camellia sinensis L.) cultivars. Hortic Res. 2018, 5, 7.

Zhao, X.M.; Wang, X.S., Du, X. Taste characteristics of purple tea leaf and reduction of bitterness and astringency. Journal of Tea Science. 2009, 28: 372-378.

Wang, J.H.; Yue, G.; Liu, S.J. Research on biochemical components of purple bud-tea and development of health beverage containing them. The Beverage Industry.2003,01:15-18.

Point 5: line 19 "has major economic values" something is missing

Response: We are grateful to the reviewer for your meticulous inspections and we have modified to “the special flavor of purple-leaf tea is highly different from that of green-leaf tea, and which is it has the major ingredients of its economic value” in line 19.

Point 6: line 20 "morphological differentiation" this is not being studied

Response: We are grateful to the reviewer for your meticulous inspections and sorry for our mistakes. We have changed to “morphological differentiation” by “phenotypic charateristics” in line 20

Point 7: line 25 clarified the spatial patterns... how?

Response: We are grateful to the reviewer for your meticulous inspections and sorry for our mistakes. We have changed to “The high expression of genes related to flavonoid biosynthesis (e.g., PAL and LAR) clarified the spatial patterns of biosynthesis and the accumulation of these metabolites” by “the high expression of genes related to flavonoid biosynthesis (e.g., PAL and LAR) exhibits the specific expression of biosynthesis and the accumulation of these metabolites” in line 25.

Point 8 line 26 I am not sure you found a correlation, explain

Response: The UGFT enzyme of plants was the final gene in the anthocyanin pathway. The transfer of the glucosyl moiety from UDP-glucose to the 3-hydroxyl group of anthocyanidins by UFGT was shown to be the key for anthocyanidin stability and water solubility (Li et al., 2017; Zhang et al., 2016). Compared with HD metabolites, our transcriptome data showed that the expression of two UFGT genes was up-regulated, while the metabolome data showed that anthocyanin was mainly glycosylated anthocyanin. Thus, we suspect that UFGT enzyme is related to anthocyanin components in purple leaf tea JMZ.

Wu, X.X; Gong, Q.H.; Ni, X.P.; Zhou, Y.; Gao, Z.H. UFGT: The Key Enzyme Associated with the Petals Variegation in Japanese Apricot. Front Plant Sci. 2017, 8: 108.

Li, X.J.; Zhang, J.Q.; Wu, Z.C.; Wu, Z.C.; Lai, B.; Huang, X.M.; Qin, Y.H.; Wang, H.C.; Hu, G.B. Functional characterization of a glucosyltransferase gene, LcUFGT1, involved in the formation of cyanidin glucoside in the pericarp of Litchi chinensis. Physiol Plant. 2016. 156: 139‐149.

Point 9: line 27 what do you mean by "co-expression analysis"?

Response: Co-expression analysis refers to the network analysis of gene co expression, which is based on the relationship between gene expression data. It can reflect the relationship between gene expression and regulation. Co-expression analysis is also used to analyze the relationship between genes and metabolites, which appears in many bionomics literatures (Liu et al., 2019; Li et al., 2019).

Liu, Y.Y.; Chen, X.R.;Wang, J.P.; et al. Transcriptomic analysis reveals flavonoid biosynthesis of Syringa oblata Lindl. In response to different light intensity. BMC Plant Biology. 2019, 19: 487.

Li, H.Y.; Lv, Q.Y.; M, C.; et al. Metabolite profiling and transcriptome analyses provide insights into the flavonoid biosynthesis in the developing seed of tartary buckwheat (Fagopyrum tataricum). Joutnal of Agricultural and Food Chemistry. 2019,67: 11262-11276.

The flavor substances of tea are mostly secondary metabolites, which are the material basis of determining the quality of tea. The secondary metabolites are related to individual characteristics and regulated by the expression of biosynthetic genes. The biosynthesis function gene regulates the accumulation of secondary metabolites, and then affects the quality of tea. In our study, we took Camellia sinensis cv. Jinmingzao (JMZ) with purple-leaf as the research object, obtained the differential gene and the differential metabolite compared with Camellia sinensis cv. Huangdan (HD) with green-leaf tea, and analyzed the correlation between the flavonoid metabolite components of JMZ and the expression of nine enzymes, so as to confirm the catalytic effect of key enzymes. It is of great value to study the synthesis of gene regulated metabolites.

Point 10: line 36 widely grown in China

Response: Thanks for the Reviewer’s kind advice, we have changed to “widely planted in China” by “widely grown in China” in line 36.

Point 11: line 39 largest secondary metabolites; something is missing

Response: We are grateful to the reviewer for your meticulous inspections and we have modified to “Flavonoids, as the largestmost pronounced secondary metabolites in plants, are associated withcontributed to various human health benefits and conditions, including such as anti-oxidant activityability, anti-aging, anti-cancer, and balance blood glucose, which could and reduced riskincidence of vascularcardivascular disease, liver injurycancer, diabetes and hypertension ” in line 39.

Point 12: line 47 figs

Response: We have modified to “figs” in line 47 according to the comment.

Point 13: line 53-55 rephrase sentensse, what do you mean by "ornamental systems"?

Response: We are grateful to the reviewer for your meticulous inspections and we have changed to “ornamental systems” by “colored-leaf plants” in line 53-55

Point 14: line78 widely grown variety

Response: We have changed to “widely planted variety” by “widely grown variety” in line 36 according to the comment. 

Point 15: line 79 "tender shoots" other places you call it young leaves, please define the growth stage and be consistent

Response: Thanks for the Reviewer’s kind advice. We have changed to “tender shoots” by “young leaves” in line 79.

Point 16: line 81 what is test samples? the "mix" in figure 2a?

Response: We have realized that the expression of test sample is unclear. We have rewrited “Samples were obtained from three plants of each cultivar as for three biological replicates. The tender samples from the same tea plant were collected and mixed together as a biological replicate” in line 81.

Point 17: line 88-89, be precise

Response: We are grateful to the reviewer for your meticulous inspections and we have removed “about” in line 88-89 according to the comment.

Point 18: line 131 technical replicates were for the analyses, please explain

Response: Real time quantitative (PCR) is a common method for gene expression analysis and verification in molecular biology experiments. To ensure the reliability and stability of qPCR results, we have performed the biological and technical differences. To control technical differences, each sample includes at least three technical repeats. The repeated reaction components of each technology are the same. Briefly, in the experiment design, each tea variety contains three biological samples, all of which are designed with three technical repetitions, with a total of nine reaction processes.

Point 19: line 156 here you call them cultivars other places varieties, be consistent

Response: Thanks for the Reviewer’s kind advice. We have modified to “varieties” by “cultivars” thought the text according to the comment.

Point 20: the legend should explain the figure wit sufficient detail, what is a HD plant etc.

Response: Thanks for the Reviewer’s kind advice. We have added to “C. sinensis cv. Jinmingzao (JMZ) and C. sinensis cv. Huangdan (HD)” in the legend.

Point 21: line 160-161 rephrase sentense

Response: Thanks for the Reviewer’s kind advice, we have rephrased sentence in line 160-161 by “To compare the metabolite changes of purple-leaf tea JMZ and green-leaf tea HD, dataset obtained from LC-ESI-MS/MS were subjected to PCA and PLS-DA analyses. The results revealed that purple-leaf tea JMZ and green-leaf tea HD clearly separated in score plot”. 

Point 22: legend to figure 2 could be more informative

Response: Thanks for the Reviewer’s kind advice. We have changed to “Multivariate statistical analysis of metabolites from C. sinensis cv. Jinmingzao (JMZ) and C. sinensis cv. Huangdan (HD). (a) PCA score plot of metabolites between the young leave of JMZ (red) and HD (green); the x-axis represents the first principal component and the y-axis represents the second principal component. (b) OPLS-DA score plot of metabolites between the young leaves of JMZ (red) and HD (green). (c) Volcano plot of metabolites between the young leaves of JMZ and HD. Metabolites with q-value >0.05 are highlighted in red for upregulation and green for downregulation; (d) The 124 SCMs were divided into 10 categories, of which 69 were upregulated.” in the legend (figure 2).

Point 23: legend to figure of JMZ and HD varieties

Response: Thanks for the Reviewer’s kind advice. We have added to “C. sinensis cv. Jinmingzao (JMZ)” and “C. sinensis cv. Huangdan (HD)” according to the comment..

Point 24: line 201 that are widely exist, rephrase

Response: We are grateful to the reviewer for your meticulous inspections and we have modified to “Phenolic acids are aromatic secondary metabolites that widely distribute in the plant kingdom” in line 201. 

Point 25: Figure 5, give reference to the pathway shown

Response: Thanks! We have added the reference according to your comment.

[38] Wei, C.L.; Yang, H.; Wang, S.B.; Zhao, J.; Liu, C.; Gao, L.P.; Xia, E.H.; Lu, Y.; Tai, Y.L.; She, G.B.; Sun, J.; Cao, H.S.; Tong, W.; Gao, Q.; Li, Y.Y.; Deng, W.W.; Jiang, X.L.; Wang, W.Z.; Chen, Q.; Zhang, S.H.; Li, H.J.; Wu, J.; Wang, P.; Li, P.H.; Shi, C.Y.; Zheng, F.Y.; Jian, J.B.; Huang, B.; Shan, D.; Shi, M.M.; Fang, C.B.; Yue, Y.; Li, F.D.; Li, D.; Wei, S.; Han, B.; Jiang, C.J.; Yin, Y.; Xia, T.; Zhang, Z.Z.; Bennetzen, J. L.; Zhao, S.C.; Wan, X.C., Draft genome sequence of Camellia sinensis var. sinensis provides insights into the evolution of the tea genome and tea quality. Proceedings of the National Academy of Sciences of the United States of America. 2018, 115: E4151-e4158 

Point 26: Figure 6 expression of XX representative genes in the YY biosynthetic pathway of ....

Response: Thanks for the Reviewer’s kind advice. We have added to “PAL, phenylalanine ammonia-lyase; C4H, cinnamic acid hydroxylase; 4CL, coumadin CoA ligase; CHS, chalcone synthase; CHI, chalcone isomerase; F3H, flavonoid 3-hydroxylase; F3’H, flavonoid 3’-hydroxylase; F3’5’H, flavonoid 3’ 5’-hydroxylase; DRF, dihydroflavonol 4-reductase; LAR, leucoanthocyanidin reductase; UFGT, UDP glucose-flavonoid 3-o-glcosyl-transferase.” in the legend of figure 6.

Point 27: line 303 This gene had corrections... I do not understand

Response: We are grateful to the reviewer for your meticulous inspections. The expression level of CsHCT was identified that it was positively related to the contents of flavonoids, dihydroflavone, isoflavones, chalcones, anthocyanins, flavanols, dihydroflavonol, and flavonoid carbonside in purple-leaf tea JMZ. Therefore, the results showed that CsHCT is a key regulator of the flavonoids biosynthesis in purple-leaf tea JMZ.

Point 28: line 307 with four and nine flavonoids, which ones?

Response: We are grateful to the reviewer for your meticulous inspections. In our study, we searched for the transcription factor that may be involved in the regulated of flavonoid biosyathesis. In total of 45 MYB TFs and 51 bHLH TFs were identified. The expression level of 45 MYB TFs highly correlated (|r| > 0.9) with four flavonoid (flavonoid carbonoside, flavanols, dihydroflavonol, and anthocyanins). Fifty-one bHLH TFs were remarkable as a high correlation with nine flavonoids (flavonoids, dihydroflavone, isoflavones, chalcones, anthocyanins, flavanols, flavonols, dihydroflavonol, and flavonoid carbonoside).

Point 29: line 322, the section on catechin content needs to be rewritten

Response: We sincerely appreciate the reviewer for constructive comments. We have rewritten to the section on catechin content in line 322.

“Catechins are responsible for astringency and bitterness [8]. The content of C, EC, EGCG, EGC, GC and ECG were identified between JMZ and HD through UPLC-TOF-MS (Table 2). The EC, EGCG, EGC, GC and ECG content were significantly high in purple-leaf tea (JMZ) than in the control leaves (HD). The results showed that the EC, EGCG, and ECG were predominant catechin compounds in JMZ, and reached 18.58, 55.17, and 22.88 mg/g, respectively. The total catechin amounts in JMZ were significantly (1.4 folds) higher than those in HD.”

Point 30: Figure 8 reference is missing

Response: Thank you! We have added the reference according to your comment.

[40] Liu, S.R.; An, Y.L.; Tong, W.; Qin, X.J.; Samarina, L.D.; Guo, R., Xia, X.B.; Wei, C.L. Characterization of genome-wide genetic variations between two varieties of tea plant (Camellia sinensis) and development of InDel markers for genetic research. BMC Genomics, 2019, 20, 1-16.

Point 31: Line 343-345 "fill the existing research gaps" how?

Response: We are grateful to the reviewer for your meticulous inspections. Although some studies have investigated the genes regulated. Although recent studies have investigated the dynamic transcriptome during tea plant development, most have focused on a single tea cultivar, so there is a lack of comparative data representing different cultivars.

Point 32: line 350 major flavonoid, rephrase

Response: Thanks for the Reviewer’s kind advice, we have rephrased sentence in line 160-161 by “Among these flavonoids, anthocyanins are the the key pigments that impart color to leaves”.

Point 33: line 358 Six main type of plants occur, rephrase

Response: Thanks for the Reviewer’s kind advice, we have rephrased sentence in line 358 by “The plant contained six anthocyanin compounds, namely, peonidin, delphinidin, petunidin and malvidin, syringes and anthocyanin”.

Point 34: line 361 Zijuan ???

Response: We are grateful to the reviewer for your meticulous inspections. We have added to“C. sinensis. cv. Zijuan” in line 361.

Point 35: line 367 absence of anthocyanin in HD, please elaborate and explain

Response: We are grateful to the reviewer for your meticulous inspections. The coloration and stability of anthocyanins are closely related to their modification. The modification methods include glycosylation, methylation and acylation. Glycosylation enhances the stability and water solubility of anthocyanin; methylation stabilizes the B ring of anthocyanin, reduces the chemical activity of the whole molecule, and increases its water solubility. Thus, we made this hypothesis about absence of anthocyanin in HD.

Yonekura-Sakakibara, K.; Nakayama, T.; Yamazaki, M.; Saito, K., Modification and Stabilization of Anthocyanins. 2009; p169-190.

Point 36: line 369-370 rephrase

Response: Thanks for the Reviewer’s kind advice, we have rephrased sentence in line 358 by “the main concentration of anthocyanin 3-O-glucoside on the cultivars of purple leaves agreed with the previous reports of Shen et al.”.

Point 37: line 374. wrong the is not report on sonsory characteristics

Response: Thanks for the Reviewer’s kind advice, we have rephrased sentence in line 358 by “phenolics are the main nutritious ingredients in the fresh leaf of the tea plant. This relationship might have resulted from result of the regulation of phenolic compounds (e.g., phenolic acids, flavonoids, and tannins)”.

Point 38: line 433 "high accumulation of anthocyanins and catechins" there is not quantification data provided for the two tea varieties

Response: In our study, the relative quantification of anthocyanin in our experiment was between two tea varieties with different leaf colors, and the absolute quantification of catechin was detected.

Thank you again for your advice and hope to learn more from you!

Reviewer 2 Report

In this manuscript, authors performed a metabolomics and transcriptomic characterization of the purple tea variety Jinmingzao (JMZ) compared to the green tea variety Huangdan (HD). A sound methodology was used to perform the analysis, although statistical analysis are required in some sections of the paper. Overall, the paper is well written with the exception of the Discussion section. This section is a big mess, so I recommend rewriting it completely.

Specific comments:

Line 18, change “conten” by “content”.

Line 28, change “help identify” by “help to identify”.

Line 53, a reference must be included for this sentence.

Line 80, change “with three biological replicates” by “from three biological replicates”.

Line 96, please check the chromatographic phases, both seems to have the same composition.

Line 103, I wonder whether raw data were used to perform de multivariate analysis. A previous normalization and/or scaling of the data is advisable when performing such an analysis, otherwise those compounds with a higher response in the analyzer would have a higher contribution to the PCA or PLSDA model.

Line 103, Partial Least-Squares Discriminant Analysis (PLS-DA).

Line 131, “were used for the analysis” instead of   “were for the analysis”.

Line 144, change “catechin identified” by “catechin was identified”.

Line 144, please provide the wave-length used to identify catechin.

Line 144, ESI ionization mode?.

Line 147, change “m/Z” by “m/z”.

Figure 1, legend of the panel B change “JMA” by “JMZ”.

Line 161, remove “was performed”.

Line 163, from a statistical point of view the volcano-plot do not report statistical differences, is just a tool to easily visualize the distribution of the data on the bases of two statistical parameters. I recommend reformulating this sentence.

Line 172, please rewrite the sentence “where the elevated…”.

Line 181, chlorogenic acid methyl ester is included in both groups.

Line 183, I wonder why authors remark a 1.45-fold increase of aminoacid and derivates when, according to M&M, they stablished a threshold to considerer a metabolite as SCM in a FC ≥ 2.

Figure 2, legend of panel d, please remove “of which 69…”.

Line 201, remove “are”.

Line 217, along the manuscripts authors refer to the metabolites FC in a natural (I guess those are the number in the line 217) or a log2 scale (Figure 2C), both are correct but please maintain consistency in the text.

Line 231, include “Biological Process” as a GO category.

Line 252, the statistical parameter described in this sentence are those used for transcriptomic analysis and not for metabolomics. Please change them.

Line 258, where are these 17 genes coming from?. Are authors referring to previous publications?. If so, a reference is needed.

Figure 5, in the figure legend authors stated that “Grids with color scales from light to dark represent FPKM values”, however on the right side a heat map is shown for metabolites for which FPKM values are senseless. However, in both panels a log2FC color scale is shown for both JMZ and HD cultivars. What conditions or genotypes are compared to calculate the FC?. There is also a lack of information of what is the meaning of the different colors of the genes labels in the biosynthetic pathway.

Line 284, authors report that there is a high correlation between data from the RNAseq analysis and qPCR. I miss a proper statistical analysis on the qPCR data to verify whether the observed differences are significant. Anyway, data showed in the Fig.6 from 11 out of the 17 genes evaluated show a different tendency that data presented in the table S3, so authors must reconsider the former assertion.

Line 295, please rewrite the sentence starting by “the results revealed”.

Table 2, statistical information is missing. What is the meaning of the asterisks present in the table?.

Author Response

Dear Reviewer:

Thank you for your comments concerning our manuscript entitled “Comparision of metabolome and transcriptome of flavonoid biosynthesis pathway in a pueple-leaf tea germplasm Jinmingzao and a green-leaf tea germplasm Huangdan reveals the relationship with genetic mechanisms of color formation”. All of your comments are valuable and very helpful for revising and improving our manuscript, and we are also appreciated that your comments have given us a lot of inspiration and different ideas for our future research. We have studied comments carefully and have made corrections. Accordingly, we have uploaded the revised manuscript with all the changes highlighted by using the track changes mode in MS Word. The corrections in the paper and the responds to the reviewer’s comments are as flowing:

Comments and Suggestions for Authors

In this manuscript, authors performed a metabolomics and transcriptomic characterization of the purple tea variety Jinmingzao (JMZ) compared to the green tea variety Huangdan (HD). A sound methodology was used to perform the analysis, although statistical analysis are required in some sections of the paper. Overall, the paper is well written with the exception of the Discussion section. This section is a big mess, so I recommend rewriting it completely.

Response:Thanks for the Reviewer’s suggestion. After thought about your suggestion, we have rewritten the Discussion section.

Point 1: Line 18, change “conten” by “content”.

Response: We are grateful to the reviewer for your meticulous inspections and we have modified to “content”" in line 18.

Point 2: Line 28, change “help identify” by “help to identify”.

Response: Thanks for the Reviewer’s kind advice. We have modified change “help identify” by “help to identify” according to the comment.

Point 3: Line 53, a reference must be included for this sentence.

Response: Thanks for the Reviewer’s kind advice. We have added a reference.

[21] Pan, Y.Y.; Wu, H.L.; Li, J.X.; Yang, C.W.; Liu, J. Advances in research and utilization of purple tea. Guangdong Agricultural Sciences. 2015, 42: 8-12+17.

Point 4: Line 80, change “with three biological replicates” by “from three biological replicates”.

Response: Thanks for the Reviewer’s kind advice, we have changed “with three biological replicates” by “from three biological replicates” in line 80 according to the comment.

Point 5: Line 96, please check the chromatographic phases, both seems to have the same composition.

Response: We are grateful to the reviewer for your meticulous inspections and sorry for our mistakes. We have modified to "The mobile phase consisted of A containing 0.04% acetic acid in deionized water" in the corresponding location.

Point 6: Line 103, I wonder whether raw data were used to perform de multivariate analysis. A previous normalization and/or scaling of the data is advisable when performing such an analysis, otherwise those compounds with a higher response in the analyzer would have a higher contribution to the PCA or PLSDA model.

Response: Thank you! We normalized and scaled the raw data, then, used the obtained results for PCA and PLS-DA analysis.

Point 7: Line 103, Partial Least-Squares Discriminant Analysis (PLS-DA).

Response: We have modified to “Partial Least-Squares Discriminant Analysis (PLS-DA)” in line 103 according to the comment.

Point 8: Line 131, “were used for the analysis” instead of “were for the analysis”.

Response: Thanks for the Reviewer’s kind advice. We have changed to “were used for the analysis” in line 131.

Point 9: Line 144, change “catechin identified” by “catechin was identified”.

Response: We have changed to “catechin was identified” in line 131 according to the comment.

Point 10: Line 144, please provide the wave-length used to identify catechin.

Response: The detection of catechin uses UPLC-TOF-MS without wavelength setting. MS adopted the MSe scanning mode, mass spectrum, scanning range of 100–1000 m/z, scanning rate of 0.1 s, centroid data format, low collision energy of 5 V, and high collision energy of 20–40 V.

Point 11: Line 144, ESI ionization mode?.

Response: Yes! We have modified to “ESI ionization mode” in line 144.

Point 12: Line 147, change “m/Z” by “m/z”.

Response: We have changed “m/Z” by “m/z” in line 147.

Point 13: Figure 1, legend of the panel B change “JMA” by “JMZ”.

Response: We have changed “JMA” by “JMZ” in the corresponding location.

Point 14: Line 161, remove “was performed”.

Response: Thanks for the Reviewer’s kind advice, we have removed “was performed” in line 161.

Point 15: Line 163, from a statistical point of view the volcano-plot do not report statistical differences, is just a tool to easily visualize the distribution of the data on the bases of two statistical parameters. I recommend reformulating this sentence.

Response: Thanks for the Reviewer’s kind advice, we have modified “the volcano plot of their metabolite contents displayed a significant difference” by “Volcano plots were generated to display the significant differences between JMZ and HD” in line 163.

Point 16: Line 172, please rewrite the sentence “where the elevated…”.

Response: Thanks for the Reviewer’s kind advice, we have rewritten as “with green representing elevated levels of metabolites and red representing decrease” in line 161.

Point 17: Line 181, chlorogenic acid methyl ester is included in both groups.

Response: We are grateful to the reviewer for your meticulous inspections and sorry for our mistakes. After checking with the raw data, we have modified to “chlorogenic acid methyl ester was decreased compare JMZ to HD.”

Point 18: Line 183, I wonder why authors remark a 1.45-fold increase of aminoacid and derivates when, according to M&M, they stablished a threshold to considerer a metabolite as SCM in a FC ≥ 2.

Response: We are grateful to the reviewer for your meticulous inspections and sorry for our mistakes. After checking, we have modified to “3.54- fold changes” in line 183.

Point 19: Figure 2, legend of panel d, please remove “of which 69…”.

Response: Thanks for the Reviewer’s kind advice, we have removed “of which 69 were upregulated” in the corresponding location.

Point 20: Line 201, remove “are”.

Response: We have removed “are” in line 201.

Point 21: Line 217, along the manuscripts authors refer to the metabolites FC in a natural (I guess those are the number in the line 217) or a log2 scale (Figure 2C), both are correct but please maintain consistency in the text.

Response: We sincerely appreciate the reviewer for constructive comments. We have modified to “15.203- and 15.114-fold changes” in line 217.

Point 22: Line 231, include “Biological Process” as a GO category.

Response: We sincerely appreciate the reviewer for constructive comments. We have added “Biological Process” in the corresponding location.

Point 23: Line 252, the statistical parameter described in this sentence are those used for transcriptomic analysis and not for metabolomics. Please change them.

Response: We sincerely appreciate the reviewer for constructive comments. We have modified to “the fold change ≥ 2 between purple and green-leaf tea (P < 0.05)” by “ratio > 2 or ratio ≤1/2, q-value ≤ 0.05 and VIP ≥1 between purple and green-leaf tea” in line 252.

Point 24: Line 258, where are these 17 genes coming from?. Are authors referring to previous publications?. If so, a reference is needed.

Response: We obtained 17 core differential genes from the JMZ transcriptome dataset together with KEGG pathway of flavonoids and related expression genes. We have added the reference in line 258.

[38] Wei, C.L.; Yang, H.; Wang, S.B.; Zhao, J.; Liu, C.; Gao, L.P.; Xia, E.H.; Lu, Y.; Tai, Y.L.; She, G.B.; Sun, J.; Cao, H.S.; Tong, W.; Gao, Q.; Li, Y.Y.; Deng, W.W.; Jiang, X.L.; Wang, W.Z.; Chen, Q.; Zhang, S.H.; Li, H.J.; Wu, J.; Wang, P.; Li, P.H.; Shi, C.Y.; Zheng, F.Y.; Jian, J.B.; Huang, B.; Shan, D.; Shi, M.M.; Fang, C.B.; Yue, Y.; Li, F.D.; Li, D.; Wei, S.; Han, B.; Jiang, C.J.; Yin, Y.; Xia, T.; Zhang, Z.Z.; Bennetzen, J. L.; Zhao, S.C.; Wan, X.C., Draft genome sequence of Camellia sinensis var. sinensis provides insights into the evolution of the tea genome and tea quality. Proceedings of the National Academy of Sciences of the United States of America. 2018, 115: E4151-e4158 

Point 25: Figure 5, in the figure legend authors stated that “Grids with color scales from light to dark represent FPKM values”, however on the right side a heat map is shown for metabolites for which FPKM values are senseless. However, in both panels a log2FC color scale is shown for both JMZ and HD cultivars. What conditions or genotypes are compared to calculate the FC?. There is also a lack of information of what is the meaning of the different colors of the genes labels in the biosynthetic pathway.

Response: We sincerely appreciate the reviewer for constructive comments. We have redrawn Figure 5, and changed the corresponding legent to “Transcript profiling of genes in the phenylpropanoid and flavonoid biosynthetic pathways in C. sinensis cv. Jinmingzao (JMZ) and C. sinensis cv. Huangdan (HD). Proposed pathway of biosynthesis derived from the literature [38]. Gene expression is displayed as heat map depicting the log2 (FPKM) values. Red and green font indicate up- and down- regulated genes, respectively. PAL, phenylalanine ammonia-lyase; C4H, cinnamic acid 4-hydroxylase; 4CL, 4-coumarate CoA ligase; CHS, chalcone synthase; CHI, chalcone isomerase; F3H,flavanone 3-hydroxylase; F3’H, flavanoid 3′-hydroxylase; DFR, dihydroflavonol flavonol synthesis; ANS/LDOX, anthocyanidin synthase/leucocyanidin oxygenase; LAR, leucocyanidin reductase; UFGT, UDP glucose-flavonoid 3-o-glcosyl-transferase.

.

Point 26: Line 284, authors report that there is a high correlation between data from the RNAseq analysis and qPCR. I miss a proper statistical analysis on the qPCR data to verify whether the observed differences are significant. Anyway, data showed in the Fig.6 from 11 out of the 17 genes evaluated show a different tendency that data presented in the table S3, so authors must reconsider the former assertion.

Response: Thanks! We have further analyzed the correlation of flavonoid related-gene from qRT-PCR and RNA-Seq results. Although the R2 value is low, the results of partail genes are good. We also found that the difference between qRT-PCR and RNA-Seq was due to the low expression of selected genes. We also realized the interest in the expressed genes can deepen the sequencing depth in experiment design. We have modified as “With respect to the purple-leaf, 6/17 (35%) were found to be inconsistent with respect to expression between the qRT-PCR and RNA-Seq data. Overall, 11/17 (65%) were consistent with respect to expression between the qRT-PCR and RNA-Seq data. Those included CsPAL, CsC4H-1, Cs4CL, Cs4CL-2, CsCHI-1, CsCHI-2, CsF3’5’H, CsF3’H, CsDFR-1, CsUFGT-1, CsUFGT-2. Detailed information of the main flavonoids-related genes up- and down-regulated are presented in Fig.6”.

Point 27: Line 295, please rewrite the sentence starting by “the results revealed”.

Response: We have rewritten as “among the detected transcription factors, 66 MYBs and 187 bHLH were identified” in line 295.

Point 28: Table 2, statistical information is missing. What is the meaning of the asterisks present in the table?.

Response: We sincerely appreciate the reviewer for constructive comments. We have added “* means significant difference, P<0.05” in legend of table 2.

Thank you again for your advice and hope to learn more from you!

Round 2

Reviewer 1 Report

The authors have address my concerns, but still something needs attention, in general what has been added shown by track-changes needs to be check for English writing and grammar.

Here are some addition remarks:

Title is better now

line 31 shows

line 32, the two CsUFGTs upregulation or downregulation?? you need to give more information, also why that is interesting in few words

line 33 "chosen" or identified?

line 42 to 43 can not be understood!

line 45-48 is now more informative, but needs correction

line 56 "in the gene" what do you mean?

line 61 other currentty ornamental plants, something is missing to complete the statement

Line 86, please include ploidy level, are they diploid or polyploids? this is important for the study of gene families. You write they are new, are you sure they homozygotes?

line 88 Wuqu, if it is located in mountains, please write the altitude

line 89-97 it is still not clear to me which samples are taken, I think you are using different terms to identify the leave in the text, secondly three samples are taken to have biological replicates, but apparently after harvest the "tender samples from the same tea plant were collected and mixed together as a biological replicate", does that mean that you harvest a number of leaves (what is tender?) from plant no 1, and store as replicate no.1 (JMZ_1) and so on for two more plants of JMZ? What is the relation between these and the 6 libraries?

In figure 1A, what is mix_1?

what is the difference tween "tender shoots", "tender" and "leave", I think you complicate things by not being strickt

line 99-100 I can guess the name of the mixer-mill, but you do not spell it correctly

line 99 zirconium (40. element)

line 123 shoots or was it leaves or was tender.... be consistent

line 143 biological samples were used

line 167 different mutant? how do you know they are mutants?

line 189 marked with green to represent...

line 315 legends to Figure 6, include information on statistics

line 342 biosynthesis

line 484 to 485, nice but still needs english check

line 526 citation by numbers

Author Response

Dear Reviewer:

Thank you for your comments concerning our manuscript entitled “Comparision of metabolome and transcriptome of flavonoid biosynthesis pathway in a pueple-leaf tea germplasm Jinmingzao and a green-leaf tea germplasm Huangdan reveals the relationship with genetic mechanisms of color formation”. All of your comments are valuable and very helpful for revising and improving our manuscript, and we are also appreciated that your comments have given us a lot of inspiration and different ideas for our future research. We have studied comments carefully and have made corrections. Accordingly, we have uploaded the revised manuscript with all the changes highlighted by using the track changes mode in MS Word.

The corrections in the revised manuscript and the responds to the reviewer’s comments are as flowing:

Point 1: line 31 shows

Response: Thanks for the Reviewer’s kind advice. We have modified to “shows” according to the comment.

Point 2: line 32, the two CsUFGTs upregulation or downregulation?? you need to give more information, also why that is interesting in few words

Response: Thanks for the Reviewer’s kind advice. We have modified to “two CsUFGTs were positively related to the accumulation of anthocyanin” in line 32.

Point 3: line 33 "chosen" or identified?

Response: We sincerely appreciate the reviewer for constructive comments. We have modified to “identified” in line 33.

Point 4: line 42 to 43 can not be understood!

Response: We have rewritten as “Tea plants are grown widely in the tropical and subtropical zones around the world, including mainly China, Japan, India, and Kenya” in line 42 to 43.

Point 5: line 45-48 is now more informative, but needs correction

Response: Thanks for the Reviewer’s kind advice. We have rewritten as “flavonoids, as the most pronounced secondary metabolites in plants, play an important role in human health benefits and have potential physiological functions with anti-oxidant ability, anti-aging, anti-cancer, and balance blood glucose. The intake of flavonoid composition in tea can reduce incidence of cardiovascular disease, cancer, diabetes and hypertension” in line 45-48.

Point 6: line 56 "in the gene" what do you mean?

Response: We are grateful to the reviewer for your meticulous inspections and we have modified to “The red color of cotton is mainly caused by flavonoid accumulation. Except for 4CL and F3’5’H of flavonoid related genes in red cotton, other genes that regulate flavonoid biosynthesis have higher transcription levels than those of white cotton” in line 56.

Point 7: line 61 other currentty ornamental plants, something is missing to complete the statement

Response: Thanks for the Reviewer’s kind advice. We have modified to “the regulatory pattern of flavonoid biosynthesis in purple-leaf tea differs from coloring mechanism of other currently ornamental plants but also does not match those characterized for regulating flavonoid in C. sinensis” in line 61

Point 8: Line 86, please include ploidy level, are they diploid or polyploids? this is important for the study of gene families. You write they are new, are you sure they homozygotes?

Response: ‘Jinmingzao’ is a diploid tea germplasm. It is a natural hybrid progeny of Jinguanyin, a cultivar approved by the National Crop Approval Committee in 2002 with number GS20020017. Its leaf color is highly different from that of Camellia sinensis cv. Jinguanyin, so it is a new germplasm

Point 9: line 88 Wuqu, if it is located in mountains, please write the altitude

Response: Thanks for the Reviewer’s kind advice. We have added to “150-200 m above sea level” in line 88

Point 10: line 89-97 it is still not clear to me which samples are taken, I think you are using different terms to identify the leave in the text, secondly three samples are taken to have biological replicates, but apparently after harvest the "tender samples from the same tea plant were collected and mixed together as a biological replicate", does that mean that you harvest a number of leaves (what is tender?) from plant no 1, and store as replicate no.1 (JMZ_1) and so on for two more plants of JMZ? What is the relation between these and the 6 libraries?

Response: Thanks for the Reviewer’s kind advice. We’ve all changed “young shoots”. As shown in the graph below, we harvest a number of young shoots (one bud and two leaves) from plant no 1, and store as replicate no.1 (JMZ_1) and so on for two more plants of JMZ (JMZ_2 and JMZ_3). JMZ and HD have three biologically repeated samples for transcriptome sequencing, respectively. Thus, there are six libraries in total.

We have modified to “For each cultivar, one bud and two young leaves were harvested from 24 individuals used for the metabolite analysis (3 biological replication × 4 individual for each replicate) and RNA-seq (3 biological replication × 4 individual for each replicate) on March 31, 2019. Samples were randomly collected from different branches of tea plants of each cultivar. All materials were frozen immediately in liquid nitrogen, and then stored at –80 °C until further analysis. The DEGs and metabolites of purple-leaf and green-leaf tea were identified by transcriptome and widely targeted metabolomics, and functional genes with significant differences were verified by qRT-PCR. Six libraries, namely, JMZ_1, JMZ_2, JMZ_3, HD_1, HD_2, and HD_3 were constructed during the experiment” in line 89-97.

Point 11: In figure 1A, what is mix_1?

what is the difference between "tender shoots", "tender" and "leave", I think you complicate things by not being strickt

Response: Thanks for the Reviewer’s suggestion. Figure 1A shows the standard samples we harvested (one bud and two leaves), which we wrote as “young shoots”. Mix_1 is a collection of multiple young shoots in a tea plant.

Point 12: line 99-100 I can guess the name of the mixer-mill, but you do not spell it correctly

Response: Thanks for the Reviewer’s kind advice. We have modified to “mixer-mill” according to the comment.

Point 13: line 99 zirconium (40. element)

Response: Thanks for the Reviewer’s kind advice. We have added to “(Zr; Z = 40)” according to the comment.

Point 14: line 123 shoots or was it leaves or was tender.... be consistent

Response: Thanks for the Reviewer’s kind advice. We’ve all rewritten it as “young shoots” in the manuscript.

Point 15: line 143 biological samples were used

Response: Thanks for the Reviewer’s kind advice, we have changed to “three independent biological samples were used for the analyses” in line 143 according to the comment.

Point 16: line 167 different mutant? how do you know they are mutants?

Response: Thanks for the Reviewer’s kind advice. We have rewritten as “Furthermore, compared with HD, the color of buds and leaves on the young shoots was different from that of the old leaves in JMZ plant. The buds and young leaves were purple, but the old leaves on the old leaves on the lower part of the tea plant were green.”.

Point 17: line 189 marked with green to represent...

Response: Thanks for the Reviewer’s kind advice. We have modified to “marked with green to represent elevated levels of metabolites and red to represent decrease” according to the comment.

Point 18: line 315 legends to Figure 6, include information on statistics

Response: Thanks for the Reviewer’s kind advice, we have changed to “The x-axis represents two different tea cultivars of ‘JMZ’ and ‘HD’, and the y-axis represents relative expression. The data represent the mean from three replicates with three biological repeats. Error bars indicate SE” in legend of Figure 6.

Point 19: line 342 biosynthesis

Response: Thanks for the Reviewer’s kind advice, we have modified to “biosynthesis” in line 342.

Point 20: line 484 to 485, nice but still needs english check

Response: Thanks for the Reviewer’s kind advice, we have modified to “The unique and delightful leaf color of tea is mediated by specific metabolic compounds and gene expression. The association of metabolite abundances and gene expression has been attracted increasingly interest in recent years” in line 484-485.

Point 21: line 526 citation by numbers

Response: Thanks for the Reviewer’s kind advice, we have added citation by [51].

Thank you again for your advice and hope to learn more from you!

Reviewer 2 Report

Authors have address major points of the manuscript, but some information is still missing.

  • The methodology used to normalize and scale metabolomics data must be described in M&M.
  • If authors did not used the UV spectrum to identify catechins, what spectrum they refer in line 156 when they write “absorption spectrum”.

Author Response

Dear Reviewer:

Thank you for your comments concerning our manuscript entitled “Comparision of metabolome and transcriptome of flavonoid biosynthesis pathway in a pueple-leaf tea germplasm Jinmingzao and a green-leaf tea germplasm Huangdan reveals the relationship with genetic mechanisms of color formation”. All of your comments are valuable and very helpful for revising and improving our manuscript, and we are also appreciated that your comments have given us a lot of inspiration and different ideas for our future research. We have studied comments carefully and have made corrections. Accordingly, we have uploaded the revised manuscript with all the changes highlighted by using the track changes mode in MS Word. The corrections in the paper and the responds to the reviewer’s comments are as flowing:

Point 1: The methodology used to normalize and scale metabolomics data must be described in M&M.

Response: Thanks for the Reviewer’s kind advice. We used Z-score normalized method and central scaling method to analyze raw metabolome data. Z-score normalization method normalized the mean and standard deviation of the original data. The processed data conforms to the normal distribution, i.e. the mean value is 0, the standard deviation is 1, and the conversion function is:

Central scaling: data mapping within the range of 0-1, and the formula is:

Li, B.; Tang, J.; Yang, Q.; Li, S.; Cui, X.J.; Li, Y.H.; Chen, Y.Z.; Xue, W.W.; Li, X.F.; Zhu, F. NOREVA: normalization and evaluation of MS-based metabolomics data. Nucleic Acids Res. 2017, 45: 162‐170.

Point 2: If authors did not used the UV spectrum to identify catechins, what spectrum they refer in line 156 when they write “absorption spectrum”.

Response: We are grateful to the reviewer for your meticulous inspections and sorry for our mistakes. We have modified to “catechin was identified by the standards” in line 156.

Thank you again for your advice and hope to learn more from you!